Fast2Vec, a modified model of FastText that enhances semantic analysis in topic evolution

http://orcid.org/0009-0009-6066-6609 Pertiwi Ayu 1 2
Azhari Azhari 2 arisn@ugm.ac.id
Mulyana Sri 2
1 Faculty of Computer Science, Universitas Dian Nuswantoro , Semarang , Indonesia
2 Department Computer Science and Electronics, Gadjah Mada University , Yogyakarta , Indonesia
Alatas Bilal
Electronic publication date: 2025 May 19
Publication date: 2025
Volume: 11
Electronic Location ID: e2862
Received 2024 Sep 27; Accepted 2025 Apr 7
Copyright: © 2025 Pertiwi et al.
Copyright year: 2025
Copyright holder: Pertiwi et al.
License: This is an open access article distributed under the terms of the Creative Commons Attribution License, which permits unrestricted use, distribution, reproduction and adaptation in any medium and for any purpose provided that it is properly attributed. For attribution, the original author(s), title, publication source (PeerJ Computer Science) and either DOI or URL of the article must be cited.
License URL: https://creativecommons.org/licenses/by/4.0/

Keywords: Fast2Vec, Dynamic topic model (DTM), Topic evolution, Word embedding, Natural language processing (NLP), Out-of-vocabulary (OOV) handling, Topic modeling, Semantic shift, Semantic similarity, Entropy-based topic evolution

Funding: Directorat of Higher Education Research and Technology of Indonesia 2024 2504/UN1/DITLI.1/PT.01.01/2024 This work was supported by the doctoral grant from the Directorat of Higher Education Research and Technology of Indonesia 2024 (Number: 2504/UN1/DITLI.1/PT.01.01/2024). The funders had no role in study design, data collection and analysis, decision to publish, or preparation of the manuscript.

==============================
Background

Topic modeling approaches, such as latent Dirichlet allocation (LDA) and its successor, the dynamic topic model (DTM), are widely used to identify specific topics by extracting words with similar frequencies from documents. However, these topics often require manual interpretation, which poses challenges in constructing semantics topic evolution, mainly when topics contain negations, synonyms, or rare terms. Neural network-based word embeddings, such as Word2vec and FastText, have advanced semantic understanding but have their limitations. Word2Vec struggles with out-of-vocabulary (OOV) words, and FastText generates suboptimal embeddings for infrequent terms.

Methods

This study introduces Fast2Vec, a novel model that integrates the semantic capabilities of Word2Vec with the subword analysis strength of FastText to enhance semantic analysis in topic modeling. The model was evaluated using research abstracts from the Science and Technology Index (SINTA) journal database and validated using twelve public word similarity benchmarks, covering diverse semantic and syntactic dimensions. Evaluation metrics include Spearman and Pearson correlation coefficients to assess the alignment with human judgments.

Results

Experimental findings demonstrated that Fast2Vec outperforms or closely matches Word2Vec and FastText across most benchmark datasets, particularly in task requiring fine-grained semantic similarity. In OOV scenarios, Fast2Vec improved semantic similarity by 39.64% compared to Word2Vec, and 6.18% compared to FastText. Even in scenarios without OOV terms, Fast2Vec achieved a 7.82% improvement over FastText and a marginal 0.087% improvement over Word2Vec. Additionally, the model effectively categorized topics into four distinct evolution patterns (diffusion, shifting, moderate fluctuations, and stability), enabling a deeper understanding of evolution topic interests and their dynamic characteristics.

Conclusion

Fast2Vec presents a robust and generalizable word embedding framework for semantic-based topic modeling. By combining the contextual sensitivity of Word2Vec with the subword flexibility of FastText, Fast2Vec effectively addresses prior limitations in handling OOV terms and semantic variation and demonstrates strong potential for boarder applications in natural language processing tasks.

Introduction

The information generated and stored in the digital era is growing. Text data from various sources, such as scientific studies, product evaluations, media content, and news, require thorough analysis to extract significant insights. Proper analysis is essential to transform raw scientific data into meaningful insights, optimize the value of the data and facilitate discoveries, hypothesis validation, and better decision-making in research. Research fields are often determined based on multiple factors, including a researcher’s academic path, societal influences, and evolving trends over time. Changes in research trends significantly impact the types of studies conducted (Lou & Meng, 2023). Semantic analysis is crucial for understanding topic evolution as it enables the tracking, discovery, and interpretation of changes in language use, terminologies, and related concepts. Hu et al. (2017) highlighted that semantic similarities between words play a vital role in topic evolution, influencing how topics are assessed. Additionally, Gao et al. (2022) emphasized that these similarities contribute to shaping semantic regularity in textual data.

Traditional topic modeling approaches, such as latent Dirichlet allocation (LDA), probabilistic latent semantic analysis (PLSA), and non-negative matrix factorization (NMF), have been effective in uncovering latent topics within large document collections (Muthusami et al., 2024). Among them, LDA has demonstrated strong performance in identifying themes from short texts; however, it struggles with temporal dynamics. To address this, the dynamic topic model (DTM) was introduced to incorporate time in topic evolution analysis (Blei & Lafferty, 2006). Despite this advancement, DTM is limited in capturing subtle semantic shifts influenced by context, negation, and synonyms variations. For instance, words with negation, such as “dislike” require distinct handling compared to positive expressions like “like,” whereas synonyms such as “good” and “excellent” necessitate precise semantic differentiation.

Embedding-based approaches, including Word2Vec and FastText, offer improved semantic representations to overcome some of these limitations. Word2Vec effectively maps words with similar meanings into adjacent vector space (Gao et al., 2022), but struggles with out-of-vocabulary (OOV) words. FastText enhances this by incorporating subword information, improving the representation of rare or unseen terms. However, it still faces challenges in modeling complex semantic relationships. Most recent models, such as bidirectional encoder representations from transformers (BERT), provide contextualized embeddings and have shown superior performance over topic modeling methods like LDA, particularly in classification tasks (Kaviani & Rahmani, 2020). However, BERT’s advantages are more pronounced in tasks prioritizing recall and precision rather than exploratory analysis aimed at identifying coherent topic distributions.

Preliminary experiments in this study indicate that BERT introduces biases due to the overrepresentation of hashtags and imbalanced token probabilities, which distorts topic modeling by emphasizing frequently occurring tokens rather than preserving nuanced semantic relationships. While BERT is effective for contextual understanding, its embedding strategy does not align with this study’s goal of capturing topic dynamics through distributional semantics rather than sentence-level contextualization.

To address these challenges, this research introduces Fast2Vec, a novel model that integrates the semantic strengths of Word2Vec with the subword modeling capability of FastText. Fast2Vec is designed to mitigate OOV issues while maintaining semantic coherence, thereby enhancing topic modeling for dynamic analysis. The primary objective of Fast2Vec is to offer a more refined understanding of topic distribution patterns across evolving semantic concepts. This study specifically investigates the following research questions: How does the Fast2Vec model provide an understanding of topic distribution patterns in various semantic concepts of topic evolution?

What changes occur in topic distribution across semantic concepts using the Fast2Vec model?

In line with the objectives, DTM was employed to determine the chronological development of themes, and Fast2Vec was employed to compute word embeddings. In this study, a set of the Science and Technology Index (SINTA) journal abstracts covering 2009 to 2023 was used to create a time series of topics. Fast2Vec generates word vectors with one hundred dimensions, and uniform manifold approximation and projection (UMAP) is used to reduce dimensionality before clustering with affinity propagation (AP). In this way, it is possible to reduce the dimensionality of words in a way that makes sense to their meanings. The research presents a novel procedure called Fast2Vec, which combines Word2Vec and FastText to cope with OOV words and improve the meaning of words incorporated in moving subject models. Initial findings suggest that this model is likely to improve upon earlier models, which tended to concentrate on the comprehension of topics while ignoring the change in the meaning of each subject over time.

The remainder of this article is structured as follows: ‘Related Works’ discusses related work and previous research. ‘Methods’ describes the methodology and experiment setup. ‘Experiments and Results’ presents the results and analysis. Finally, the ‘Conclusion’ concludes the study and outlines future directions for research.

Related works

Topic model for topic evolution

Topic models have been widely applied across various disciplines to analyze diverse text types, ranging from short texts to extensive documents. Initially defined by Blei, Griffiths & Jordan (2010), topic modeling is a statistical method that identifies latent semantic structures and underlying topics within large datasets (Churchill & Singh, 2022). Further, topic modeling is an unsupervised mathematical model that processes document collections to generate topics that accurately and coherently represent their content. Researchers have employed topic models in various domains, including the analysis of climate change literature (Sleeman et al., 2017), precision medicine through genomic and diabetes literature (Ni Ki et al., 2022), and strategy discussions related to global issues like the war in Ukraine (Maathuis & Kerkhof, 2023). Additionally, topic modeling has been instrumental in detecting fraud (Sánchez & Urquiza, 2024) and mapping research trends (Lou & Meng, 2023; Hu et al., 2020; Chen et al., 2020).

LDA, one of the most widely used topic modeling methods, conceptualizes topics as probability distributions over words and documents as distributions over topics (Blei, Ng & Jordan, 2003). LDA identifies thematic structures by analyzing word co-occurrence patterns, making it applicable to various natural language processing tasks (Churchill & Singh, 2022). However, LDA faces limitations in addressing temporal dynamics. To overcome this, Blei & Lafferty (2006) introduced the DTM, which incorporates time-based elements to track topic evolution. DTM has been utilized to analyze scientific entities, identify research trends, and categorize them into mainstream, short-term, and long-term directions (Lou & Meng, 2023). Applications of DTM extend to geographical planning and policy (Yao & Wang, 2020; Jiang et al., 2022) and exploring online communities engaged in abortion debates (Pleasants et al., 2023). Despite these advances, both LDA and DTM struggle with capturing semantic regularity and contextual shifts, as Kai et al. (2019) and Geeganage, Xu & Li (2021) have noted. This limitation underscores the importance of semantic analysis in understanding topic evolution, particularly in tracking changes in contextual meaning over time (Hu et al., 2017; Li et al., 2019; Chen et al., 2017).

Semantic analysis in topic evolution

Recent advancements in computational semantic technologies have driven the development of word-embedding techniques, enabling more nuanced analysis of meaning changes at the word level. Word2Vec, for instance, facilitates the exploration of semantic regularity in documents by analyzing word relationships based on contextual co-occurrence frequencies (Gao et al., 2022). Researchers have leveraged Word2Vec to uncover semantic patterns, such as Hamilton, Leskovec & Jurafsky (2018), who demonstrated how historical changes could be captured by comparing word embeddings. Similarly, Geeganage, Xu & Li (2021) utilized semantic patterns to interpret textual meaning, while Aboelela, Gad & Ismail (2021) applied semantic-based opinion-mining techniques to support decision-making.

Despite its utility, Word2Vec exhibits significant limitations, particularly in handling OOV words due to its reliance on fixed vocabularies and lack of morphological processing (Choi & Lee, 2020). To address these challenges, Bojanowski et al. (2017) proposed FastText, which extends Word2Vec by incorporating subword information through n-gram character representations, enabling improved handling of OOV words and morphological variation. FastText has been shown to enhance the understanding of word prefixes and suffixes (Athiwaratkun, Wilson & Anandkumar, 2018). It also excels in syntactic analysis, particularly in language with complex morphologies (Selva Birunda & Kanniga Devi, 2021). Asudani, Nagwani & Singh (2023) further highlighted that FastText, developed by the Facebook AI Research (FAIR) lab effectively generates vectors for unknown words based on their morphology. Furthermore, Zhang, Gao & Yuan Fang (2020) demonstrated the applicability of FastText in extending LDA to shorter sentences, improving semantic regularity in text analysis.

Most of the latest approaches in representation learning are embodied in BERT, which provides context-solid embeddings that distinguish both directional meanings (George & Sumathy, 2023). BERT can successfully be employed as a practical feature extraction technique, and a more recent work, BERT EmHash, employed BERT within the EmHash framework for hashtag recommendation tasks, demonstrating its effectiveness. However, this performance is optimal when executing the necessary classification set of tasks instead of the exploratory or topic distribution set of functions. Such encoding by BERT may introduce distortions such as excessive representations of hashtags or over-representation of frequently used terms, which would undermine the necessary semantic coherence in topic modeling and study of the evolution of the topic (Kaviani & Rahmani, 2020). It is widely recognized that BERT performs much better in tasks that do not require substantial semantic knowledge; however, its use in the context of topic dynamics remains modest.

Fast2Vec and the LDA integration model

Previous research highlighted the importance of combining topic models with word embedding techniques. Niu & Dai (2015) introduced Topic2Vec, which integrates LDA and Word2Vec to represent topics in a semantic vector space. Zhang, Gao & Yuan Fang (2020) also combined LDA with similar methods to enhance topic coherence by utilizing external corpus information using cosine distance to measure text-topic similarities in the semantic space. This approach has shown superior performance in topic categorization compared to other methods.

The Fast2Vec model integrates the FastText and Word2Vec algorithms, merging their strengths in word representation. FastText captures sub-word information, while Word2Vec generates vectors that capture semantic relationships. Fast2Vec is further integrated with LDA to improve topic analysis by identifying hidden document structures. The Fast2Vec algorithm consists of three main stages: FastText training, Word2Vec training, and merging the models. FastText training involves initializing subwords and words, calculating their weights, and using the Continuous Bag of Words (CBoW) model with negative sampling to predict target words, followed by weight updates via gradient calculations. Word2Vec training follows the same process but focuses solely on words without subwords. The final stage merges the vector matrices from FastText and Word2Vec.

Integrating Fast2Vec with LDA provides several advantages, such as leveraging sub-word information and word context, leading to more accurate word representations and improved performance in identifying hidden topics. Fast2Vec also captures rich linguistic variations, mainly through morphological analysis, making it especially valuable for languages with complex morphological structures.

Using the SINTA text corpus for experiments showed that the Fast2Vec model with LDA finds more meaningful and cohesive topics than the Word2Vec or FastText vectors alone. This study shows that the Fast2Vec model can offer researchers new insights into topic development and enhance the integration of topic models with word embeddings for analyzing topic evolution. This study aims to bridge the research gap in semantic topic analysis by capturing variations in semantic distribution over time and contributing to understanding topic evolution in scientific literature.

Methods

Dataset

The dataset for this study was extracted from scientific articles published in reputable journals in SINTA between 2009 and 2023. Each document contains key metadata, including the abstract (in English), the author’s name, title, and publication year. A rigorous preprocessing workflow was implemented to prepare the data for topic analysis. Initially, the dataset was comprised of 89,345 abstracts, but after applying filters to ensure quality and relevance, only 54,689 abstracts were retained. The selected articles’ abstracts, titles, publication years, and keywords were extracted to build the corpus. The abstracts selected were those written in English. The proposed combination models were used to capture the evolution of dynamic topics, identify the semantic concepts in documents, and then analyze the semantic distribution of topics and track changes in their characteristics over time. Table 1 shows a sample of the dataset.

Table 1 Sample dataset (Brata et al., 2023; Hailan, Albaker & Alwan, 2023).

Year	Title	Author	Abstract	
2023	Virtual reality eye exercises application based on bates method: a preliminary study	Komang Candra Brata, Hanifah Muslimah Az-Zahra, Adam Hendra Brata, Lutfi Fanani	In many situations, ophthalmologists prescribe glasses and contact lenses as a treatment method for Myopia. The other way to cure myopia is by performing Lasik surgery or using advanced tools which takes a large amount of financial effort. Dr. William Horatio Bates proposed a promising method to cure myopia with an eye exercise approach, providing opportunities to enhance the availability of low-cost eye treatment that is not easily achievable with surgery and other expensive approaches. However, many myopia patients are still unaware of this method and have no idea how to use it in their daily lives. The rapid growth of virtual reality (VR) technology has shown that this technology offers an affordable and effective solution to provide better health services. This article expands the result of an initial experiment aimed to validate the utilization of mobile VR technology to enhance the possibility of the Bates method implementation as a low-cost daily eye exercise tool. A direct diary study with 14 participants was carried out to explicitly investigate the feasibility of the proposed app. The result reveals that the proposed app could assist people with myopia when doing eye exercises and the result is quite promising in reducing the refractive levels with 0.13 diopter on average within 5 weeks period.	
2023	Transformation to a smart factory using NodeMCU with Blynk platform	Maryam Abdulhakeem Hailan, Baraa Munqith Albaker, Muwafaq Shyaa Alwan	Incorporating internet of things (IoT) in industrial systems prompted the development of industrial internet of things (IIoT) systems, which in turn enable the automation of intelligent devices to gather, analyze, and transmit data from industrial systems in real-time. This article develops a low-cost and smart industrial remote monitoring and control system based on NodeMCU microcontrollers and Blynk server platform. It is deployed to remotely monitor manufacturer’s environment and industrial equipment and control them autonomously. Also, it protects the manufacturer’s employees from fire catastrophe by warning them using a buzzer and notification. The system comprises two main parts, sensing and actuation. The sensing part consists of three subsystems that measure temperature and humidity, water flow, and flame. The actuation part consists of a water pump, light, and fan. A powerful user interface is developed based on the Blynk platform. The proposed system controls the water pump by sensing water flow autonomously. In addition, based on a fire detected, a protection system is implemented to shut down the electricity from load in case of fire event occurs. Several testing scenarios were carried on to check the response of the system, and the result shows successful implementation of the proposal to handle different situations.	

Preprocessing

One of the main challenges in preparing the dataset was the variation in journal templates across SINTA, which required significant effort to standardize the data. These variations led to inconsistencies in how metadata such as the author, title, and abstract were presented, necessitating additional steps to ensure uniformity. For instance, mixed or unnecessary information in the author column was cleaned to retain only the essential data. The preprocessing ensured that the final dataset was structured, consistent, and free from duplicates, enabling accurate and meaningful topic analysis. This thorough approach highlights the critical importance of data cleaning and standardization when working with diverse sources to ensure the reliability of subsequent studies.

Key preprocessing steps included removing entries with missing values in critical columns such as the abstract, title, year, or author information. Only abstracts written in English were retained to maintain consistency. Each abstract was further processed by tokenizing the text into individual words, applying lemmatization to normalize words to their base forms (e.g., “running” to “run”), and removing irrelevant elements such as stopwords, punctuation, numbers, and special characters. Words appearing fewer than five times were excluded, resulting in a clean and structured dataset suitable for reliable topic extraction.

Fast2Vec model

This section explains that Fast2vec, a modification of FastText, enhances semantic analysis in the evolution of the topic. This model was used to extract abstractions from scientific articles published in reputable journals in SINTA. The main stages of the study included topic detection, semantic analysis, and analysis of evolutionary development.

The graphical model proceeded through several steps. First, abstracts, titles, publication years, and keywords from the selected articles were extracted to build the corpus. The abstracts selected were those written in English. Second, DTM was used to capture the evolution of dynamic topics. Third, the Fast2Vec model, a combination of FastText and Word2vec, was employed to identify the semantic concepts in documents from 2009 to 2023. This helped analyze the semantic distribution of topics and track changes in their characteristics over time.

Word embedding was crucial in this study, as DTM alone cannot capture semantic shifts in topic evolution (Gao et al., 2022). To address this, the cosine distance between word vectors was computed to detect changes in the semantic distribution and dynamics of evolving topics. Figure 1 shows the general model diagram describing a five-stage framework for analyzing topic evolution using semantic and clustering approaches.

Figure 1 System model.

Workflow diagram of topic and semantic analysis using SINTA journal data, including preprocessing, topic detection, semantic analysis, analysis of evolving dynamic, and visualization output.

First, preprocessing involved activities such as document extraction, relevant data selection, and the application of preprocessing methods to get the text data ready. The next step, topic detection, included document classification, preparation of document-topic distribution vectors, examination of the changes in topic intensity over time, and preparation of topic movement graphs. In the third phase, the semantic analysis of the framework employed Fast2Vec for searching modified word vectors, UMAP for dimensionality reduction (McInnes, Healy & Melville, 2018), and AP clustering for constructing semantic spaces. The fourth stage, analysis of evolving dynamics, provided the estimates of the similarity of single words, topic change, topic entropy, and rate of change to identify important developments through time. The last step was the output phase, which included different illustrations, including but not limited to graphs and animations, to address the comprehensiveness of the changes in the topics concerned. This model integrates modern semantic methods and methods of clustering as a means of adequate consideration of the dynamics of topics.

Topic detection

DTM is an extension of LDA, which allows it to track topic changes over time by partitioning documents based on time slices. DTM assumes that the topic at time t-1 is the starting point for the topic at time t before changing. The model uses the Gaussian noise component and applies a spatial model that connects the natural parameters of each topic through a state space model, which describes the natural distribution of the topic βk,wt at time t. The distribution of the document topic is generated using a normal distribution with an average of α.

This study divides the period from 2009 to 2023 into 15-time intervals. The perplexity index is used as an evaluation metric to determine the optimal number of topics, according to research by Chen et al. (2017). The perplexity metric typically identifies an optimal number of topics ranging between 80 and 100. However, this number was too large for this study, which could lead to overfitting, which occurs when a model tends to capture too many random variations in the data instead of significant patterns. In addition, having too large of a number of topics can make it challenging to interpret the analysis’s results.

Before experimenting, prior discussion helped the research team determine the appropriate number of topics. First, the corpus was summarized using an LDA mode, with topic numbers between 5 and 20. The second stage involved applying DTM to optimize parameter selection, estimating around fifteen topics. However, further reassessment by domain experts was conducted to refine these results.

As shown in Table 2, the perplexity and coherence scores were analyzed for different numbers of topics, revealing that eight topics provided the best balance between coherence and model performance (perplexity: 3,030.99, coherence: 0.75), compared to other topic numbers such as five topics (perplexity: 3,250.12, coherence: 0.42) or 15 topics (perplexity: 3,120.56, coherence: 0.60).

Table 2 Comparison of model performance for different numbers of topics (n_components).

n_components	Perplexity score	Coherence score	
5	3,250.12	0.42	
8	3,030.99	0.75	
10	3,045.87	0.68	
12	3,080.45	0.65	
15	3,120.56	0.60	

By combining quantitative evaluation (perplexity and coherence scores) with qualitative assessment from domain experts, we determined that eight topics provided the most interpretable and meaningful summarization of the corpus.

Building on these findings, Table 3 presents the document-topic distribution per period as generated by DTM. Each document is associated with a probabilistic distribution across eight topics, illustrating how thematic structure evolves over time. The probability distribution θd,kt represents the strength of a document’s association with a given topic k at time t.

Table 3 Sample result of document-topic per periode.

	Topic 1	Topic 2	…	Topic 8	
Document 1	θd1,k1t=2009	θd1,k2t=2009	…	θd1,k8t=2009	
Document 2	θd2,k1t=2009	θd2,k2t=2009		θd2,k8t=2009	
…	…	…	…	…	
Document n	θdn,k1t=2022	θdn,k2t=2022	…	θdn,k8t=2022	
…	…	…	…	…	
Document 54,689	θd54,689,k1t=2023	θd54,689,k2t=2023	…	θd54,689,k8t=2023	

By analyzing these distributions, we can observe shifts in dominant topics and how their prevalence changes across different time periods. This analysis is essential for identifying emerging research trends and tracking the evolution of semantic themes within the dataset.

Determining the number of topics

We compared the Word2Vec and FastText models to evaluate the model’s performance by combining it with LDA. We used cosine distance to measure and explain similarities between documents and topics in semantic space. The results showed that FastText can handle OOV well but is less optimal in semantic analysis compared to Word2Vec. This can be seen after we eliminated OOV cases for a more balanced result. Based on these findings, we propose a new model that modifies FastText by adding weights to improve semantic representation, which we call Fast2Vec.

Suppose a word sequence. w1n consists of n words, represented as w1,w2,…,wn. The probability of observing this sequence ,P(w1n) is computed as:

(1) P(w1n)=∏t=1n⁡P(w1t−1)

where w1n={w1,w2,….,wn}, wt is the t-th word in the sequence, w1t−1 is the context consisting of all preceding words {w1,w2,….,wt−1}, and P(w1t−1) denotes the conditional probability of wt given the context.

Topic intensity and distribution probability

Table 4 presents the topic word probability distribution and illustrates the relationship between topics and words at different periods. Each row corresponds to a specific topic (e.g., Topic 1 to Topic 8), while each column represents words within the corpus (e.g., Word 1 to Word n). The probability βk,wt indicates the likelihood of a topic k being associated with the word w at time t. The probability of distribution of all words for a given topic, as well as all topics for a given document, sums to one, are expressed in Eq. (1). This table provides valuable insights into the lexical composition of topics and their evolution over time to understand how certain words define or contribute to each topic across temporal dimensions.

Table 4 Topic word probability distribution.

Year	Topic	Word 1	Word 2		Word n	
2009	Topic 1	βk1,w1t=2009	βk1,w2t=2009	….	βk1,wnt=2009	
Topic 2	βk2,w1t=2009	βk2,w2t=2009	….	βk2,wnt=2009	
…	…	…	…	…	
Topic 8	βk8,w1t=2009	βk8,w2t=2009		βk8,wnt=2009	
2010	Topic 1	βk1,w1t=2010	βk1,w2t=2010	…	βk1,wnt=2010	
Topic 2	βk2w1t=2010	βk2,w2t=2010	…	βkn,wnt=2010	
…	…	…	…	…	
Topic 8	βk8,w1t=2022	βk8,w2t=2022	…	βk8,wnt=2022	
…						
….	…	…	…	…	…	
2023	Topic 1	βk1,w1t=2023	βk1,w2t=2023	…	βk1,wnt=2023	
Topic 2	βk2,w1t=2023	βk2,w2t=2023	…	βk2,wnt=2023	
…	…	…	…	…	
Topic 8	βk8,w1t=2023	βk8,w2t=2023	…	βk8,wnt=2023	

Table 5 displays the topic intensity values, which are used to quantify the popularity of each topic over time. A document-topic probability distribution is employed to identify trend patterns, while topic probability distributions facilitate semantic analysis. The intensity of a topic is calculated as the sum of the normalized probability distribution of topic intensity, represented by ∑d=1d=M⁡θd,kt where M is the total number of documents, as shown in Eq. (2):

(2) θtk∗=∑d=1d=M⁡θd,ktM.

Table 5 Topic intensity value.

Topic	Cluster 1	Cluster 2	…	Cluster 10	
Topic 1	θk1,c1t=2009	θk1,c2t=2009	….	θk1,c10t=2009	
…	….	….	….	….	
Topic 8	θk8,c1t=2009	θk8,c2t=2009		θk8,c10t=2009	
…	….	….	….	….	
Topic 1	θk1,c1t=2022	θk1,c2t=2022	…	θk1,c10t=2022	
…	…	…	…	…	
Topic 8	θk8,c1t=2022	θk8,c2t=2022	…	θk8,c10t=2022	
Topic 1	θk1,c1t=2023	θk1,c2t=2023	…	θk1,c10t=2023	
…	…	…	…	…	
Topic 8	θk8,c1t=2023	θk8,c2t=2023	…	θk8,c10t=2023	

This formula computes the aggregate probability distribution of topic k across all documents at time t, normalized by dividing it by the M total number of documents. The values in Table 5 represent the resulting topic intensity for various clusters (e.g., Cluster 1 to Cluster 10) and topics (e.g., Topic 1 to Topic 8), providing insights into the temporal trends and variations in topic popularity over several years.

Building a semantic analysis model

The semantic analysis model aims to group words with similar semantic concepts in a two-dimensional space. Before the model training stage, a word embedding model was developed using FastText and Word2vec. The Fast2Vec algorithm produces a more comprehensive representation of vectors.

Suppose a document corpus consists of D={d1,d2,…,dN} N documents. The process begins with the initialization of the parameters α,β,γword,γsubword,d,e,w. Where α and β are the merging weights for Word2Vec and FastText. The word and subword merging weights for the FastText model are γword+γsubword=1, with the embedding dimension d, the number of iterations e, and the window size w.

Fast2Vec (modified word vector search)

The next stage is preprocessing and tokenization for words and subwords. Documents are tokenized into words and subwords. Subwords are generated from n-gram characters with a range (nmin, nmax), which helps represent OOV words more effectively. For instance, the word “word” is tokenized into <wo, wor, ord, rd>. The tokenization steps are defined as follows: S_w=TokenizeWords(D), and S_s=TokenizeSubwords(Sword).

The training continued with Word2Vec, using the CBoW model, where the embedding for the word w is predicted from the words in the context within the window w.

Suppose the target word is wt with its context consisting of the words, wt−k,…,wt−1,wt+1,…,wt+k within a window of size k. The target word wt is the word whose embedding is being predicted. In the CBOW model, the embedding of wt is calculated based on the words surrounding it within the context window. Context words, defined by the window size k, are the words before (wt−k,…,wt−1) and after (wt+1,…,wt+k) the target word. The total number of context words is 2k, with k words on each wt size. For example, consider the sentence:

“Discover hidden semantic topics within large collections of textual data.”

If wt = topics (the target word) and k = 2: context words before wt−2 = hidden, wt−1 = semantics, context words after wt+1 = within, wt+2 = large. Using these context words ( wt−2,…,wt−1,wt+1,…,wt+2), the CBOW model computes the average embedding to predict the embedding of wt = topics. A window of the CBOW equation to predict the embedding of w2v(wt) from the context words is defined in Eq. (3):

(3) w2v(wt)=W.(12k∑i=t−k,i≠tt+k⁡w2v(wi)).

W is the weight matrix used to transform the aggregated context embedding into the target word embedding.

Negative sampling improves Word2Vec’s efficiency. Instead of updating all the words in the vocabulary, only the target word wt and some negative words wneg are updated. The loss function for negative sampling is, with Eq. (4):

(4) L(wt,wcontext,wneg)=−logP(wcontext)−∑wneg⁡log(1−P(wcontext)).

P(wcontext) represents the probability of the target word wt appearing in its given context, while P(wcontext). The probability of a negative word appearing in the context is expected to be small. The goal is to maximize P(wcontext) for the target word and minimize P(wcontext) for negative words.

Training Word2Vec embedding update

Based on the loss function, the embedding update is done with stochastic gradient descent (SGD). For the wt target word, the update is conducted as follows in Eq. (5):

(5) w2v(wt)←w2v(wt)−η∂L∂w2v(wt)

where: w2v(wt) denotes the embedding vector of the target word wt.

∂L∂w2v(wt) is the gradient of the loss function L to w2v(wt)

η is the learning rate, which controls the magnitude of the update to the embedding.

Similarly, for the negative word wneg, the update is performed as follows (Eq. (6)):

(6) w2v(wneg)←w2v(wneg)−η∂L∂w2v(wneg)

where: w2v(wneg) represents the embedding vector of the negative word wneg.

∂L∂w2v(wneg) is a gradient of the loss function L to w2v(wneg).

η is the learning rate that regulates many changes to the embedding in each update.

Training in FastText

FastText uses the word ft(w) embedding and the n-gram subword ft(si) to represent a word. The combined embedding for a w word is calculated as the weighted sum of the embedding of the word and subword in Eq. (7):

(7) ft(w)=γword.fy(wword)+γsubword.(1n∑i=1n⁡ft(si))

where: γword and γsubword is the weight for the word and subword.

n is the number of subwords.

Integrating Word2Vec and FastText embedding

The integration between the embedding results from Word2Vec and from FastText is done by summing and weighting. This is done to get the FastText subword and the semantics of the Word2Vec. When the word is not in the corpus, an approach from FastText will be used. Equation (8) is defined as the sum of two embedded with weights α and β:

(8) Ew=α.w2v(w)+β.ft(w)

were set to 0.5 each, based on the weighted sum approach. This equal weighting ensures that Word2Vec and FastText contributed equally to the final embedding representation. The choice of α = 0.5 and β = 0.5 reflects the complementary strengths of the two models: Word2Vec effectively captures semantic relationships through word co-occurrence, while FastText incorporates subword-level information, enabling better handling of rare and OOV words. The best results were achieved with α = 0.5 and β = 0.5, leveraging the strengths of both models to produce more prosperous and flexible word representations. This integration has shown satisfactory performance in experiments conducted on the validation dataset and demonstrates its capability to handle various linguistic phenomena.

OOV handling

If the word w is not present in the Word2Vec vocabulary, then we only use the embedding of FastText, Ew=ft(w). Conversely, if the word w is absent in FastText, we use the embedding from Word2Vec, Ew=w2v(w). However, this condition is rare because FastText uses n-grams of subwords to handle words not in the vocabulary. Another case is when the word w is not present in both models, and we initialize the embedding with a vector of 0, i.e., Ew=0.

After training, the embedding vector must be updated based on loss function optimization techniques, such as negative sampling. Two models make updates. The first is negative sampling on Word2Vec. For the target word wt and the context of wcontext, as well as the example of negative wneg, the embedding update is done by decreasing the loss function L using SGD:

(9) w2v(wt)←w2v(wt)−η∂L∂w2v(wt)

(10) w2v(wneg)←w2v(wneg)−η∂L∂w2v(wneg)

In FastText, the word and subword embedding update is for the word w with n-gram subwords s1, s2, …, sn. Update embedding the word w:

(11) ft(w)←ft(w)−η∂L∂ft(w),

The update of the embedding of the subword si is:

(12) ft(si)←ft(si)−η∂L∂ft(si).

Once the Ew combined embedding is formed, the next step is to use this embedding to evaluate how well the embedding can capture the semantic relationships between words. Suppose we have two words, w1, and w2, then the combined embedding is Ew1 and Ew2, respectively. The following equation calculates cosine similarity:

(13) CosSim(Ew1,Ew2)=Ew1.Ew2‖Ew1‖‖Ew2‖,

where Ew1⋅Ew2 is the dot product of both embeddings, and ‖Ew1‖,and‖Ew2‖ are the norm of the embedding.

Design of semantic analysis algorithms

Algorithm 1 is responsible for organizing the entirety of the Fast2Vec algorithm. This function performs a series of critical steps, from text data preprocessing, Word2Vec, and FastText model training to creating a combined embedding and evaluating the embedding. Algorithm 2 is tasked with training two embedding models, Word2Vec and FastText. The steps in this algorithm include initializing both models, training each model with the processed data, and finally combining the training results into a directory containing both models.

Algorithm 1 Main function of the Fast2Vec model.

Input:	
data (reviews dataset of text documents), vs (embedding dimension, vector size),	
ws (window size), mc (min count, minimum frequency for a word to be included),	
ep (epochs), α (weight for Word2Vec in the embedding combination), β (weight for FastText),	
γword (weight for word in FastText), γsubword (weight for subword in FastText)	
Output: res (Evaluation results of the embeddings)	
1: S_w← TokenizeWords(data) # Preprocess words	
2: S_s← TokenizeSubwords( S_w) # Preprocess subwords	
3: models ← TrainModels( S_w, S_s, vs, ws, mc, ep, γword, γsubword) # Train Word2Vec and FastText	
4: E ← CreateEmbeddings( S_w, models, α, β) # Create combined embeddings	
5: res ← EvaluateEmbeddings(E) # Evaluate embeddings	
6: return res	

Algorithm 2 Train Word2Vec and FastText models.

Input:	
S_w (Preprocessed words), S_s (Preprocessed subwords), vs. (Model training parameters),	
ws (window size), mc (min count, minimum frequency for a word to be included),	
ep (epochs), α (weight for Word2Vec in the embedding combination)	
γword (weight for word in FastText), γsubword (weight for subword in FastText)	
Output: models (Dictionary containing trained Word2Vec and FastText models)	
1: w2v_model← Word2VecCBOW() # Initialize Word2Vec model	
2: w2v_model.train(S_w,vs,ws,mc,ep) # Train Word2Vec model	
3: ft_model← FastText() # Initialize FastText model	
4: ft_model.train(S_w,S_s,vs,ws,mc,ep,γword,γsubword) # Train FastText model	
5: models ← {“w2v” w2v_model, “ft”: ft_model} # Combine models into a dictionary	
6: Return models	

Algorithm 3 Create embeddings.

Input:	
S_w (Preprocessed words), models (Dic. containing trained Word2Vec and FastText,	
α (weight for Word2Vec in the embedding combination), β (weight for FastText),	
Output: E (Combined embeddings for all words)	
1:   E← {} # Initialize embeddings dictionary	
2:  for each sent in S_w do	
3:   for each word in sent do	
4:    w2v_model ← models[“w2v”]	
5:    ft_model ← models[“ft”]	
6:    # Ternary conditional style for inline if-else	
7:    w2v_vec ← w2v_model.get_vec(word) if word ∈ w2v_model.vocab else Zeros(vs)	
8.    # Get Word2Vec embedding	
8:    ft_vec ← ft_model.get_vec(word) if word ∈ ft_model.vocab else Zeros(vs)	
9.    # Get FastText embedding	
10:     E[word] ← α * w2v_vec + β * ft_vec # Combine embeddings with weighted sum	
11:   end for	
12:  end for	
13:  Return E	

The CreateEmbeddings algorithm aims to create combined embeddings of two models, Word2Vec and FastText, by assigning weights ( α to the first and β to the latter). The algorithm initializes the variable E as a blank dictionary (line 1), which will hold the embeddings for different words in the sentences within the set S_w obtained from pre-processed data. It then starts looping through each sentence (line 2) and each word in the sentence (line 3). Word2Vec and FastText (also in the models’ dictionary) are called up in lines 4–5 for each word. The algorithms first try to construct embeddings for the word by checking if it is in the vocabularies of each of the models; if not, a zero vector is used as a fallback, as no embedding for that word will be helpful (lines 7–8). The combined embeddings are obtained by adding weights on a linear formula and are stored in the E dictionary (line 10). This continues for every word and every sentence (lines 11–12), and ultimately, the first combined embedding E is achieved and returned by the algorithm at line 13.

Algorithm 4 Evaluate embeddings.

Input: E (Combined embeddings for all words)	
Output: res (Evaluation results, cosine similarity for word pairs)	
1:  res ← {} # Initialize results dictionary	
2:  wpairs ← GetWordPairs() # Get word pairs for evaluation	
3:  for each (w1, w2) in wpairs do	
4:   dot_prod ← Dot(E[w1], E[w2]) # Compute dot product of the embeddings	
5:   norm_w1 ← Norm(E[w1]) # Compute the norm of the first word embedding	
6:   norm_w2 ← Norm(E[w2]) # Compute the norm of the second word embedding	
7:   sim ← dot_prod/(norm_w1 * norm_w2) # Calculate cosine similarity	
8:   res[(w1, w2)] ← sim # Store similarity result	
9:  end for	
10:  Return res	

The Evaluate Embeddings algorithm can obtain the cosine similarity of word pairs using their combined embeddings by starting a loop where they get word pairs through the GetWordPairs() function and declaring an empty dictionary res to place the results. In this case, the loop looks through every word pair (w1, w2), and for the given word pair, the algorithm computes the dot product through the embeddings stored in the dictionary E and the norm (magnitude) of each word embedding. The cosine similarity is given by the dot product over the product of the norms, and this score is saved in res as the key to the pair of words during the given step. All of the above steps are done for all pairs of the given words, and the final scores for cosine similarity are saved back in res.

Word pairs were chosen based on their semantic association and the frequency of their occurrence within the corpus to assess the embedding quality. This was done with more recurrent pairs of words, such as “student” and “teacher,” to examine the model’s ability to capture general semantic concepts. Domain-specific pairs, such as “network” and “node,” in this case, tested the effectiveness of the embeddings at representing contextually specific terms. In contrast, word pairs, in general, examine the extent to which models could distinguish even ancillary terms.

The Fast2Vec method incorporated the advantages of both FastText and Word2Vec since it combined subword information with an underlying relationship between words in a context. Hence, a more accurate and meaningful representation of words was achieved. Such a hybrid method broadens the horizons for semantic regularity capture, especially when complex linguistic structures and OOV words are involved. Using fastening also provides an avenue for a more complex analysis of the changes in meanings of topics across periods using dimensionality reduction and semantic clustering techniques.

Table 6 presents the cosine similarity scores obtained from three different word embedding models: Word2Vec, FastText, and Fast2Vec. The tables are structured into two sections: Part (a) includes cases involving OOV terms, while part (b) excludes them. The similarity scores are computed using cosine similarity, which measures the angular distance between word vectors in the embedding space. Each value in the table represents the similarity score of a single word pair, computed directly from the respective embedding space, without any aggregation across multiple word pairs. The results highlight the superior performance of Fast2Vec, which achieved the highest average similarity score of 0.961 surpassing both FastText (0.945) and Word2Vec (0.793). The effectiveness of Fast2Vec is particularly evident in words such as ‘student,’ computer,’ and ‘artificial’, which exhibit stronger semantic consistency. Although Word2Vec performs well in general word representation, it faces significant challenges with OOV cases, as reflected in near-zero similarity scores for words like ‘learn,’ ‘for,’ and ‘research’. While Word2Vec outperforms FastText in domain-specific terminology, it underperforms when handling general terms. These findings further reinforce Fast2Vec ability to establish robust semantic relationships, demonstrating superior adaptability across diverse linguistic contexts compared to its counterparts.

Table 6 Similarity test OOV cases.

Target words	Cosine similarity	
Word2vec	FastText	Fast2Vec	
Similarity test with out-of-vocabulary cases	
Student	0.972235	0.969822	0.980334	
Network	0.960879	0.900816	0.949458	
Computer	0.926148	0.876713	0.976250	
Medical	0.940626	0.941894	0.953158	
School	0.955998	0.940223	0.970204	
Architecture	0.940479	0.946469	0.973333	
Artificial	0.874789	0.936170	0.964170	
Science	0.891511	0.935207	0.970363	
Engineering	0.909928	0.904404	0.958697	
Academic	0.957779	0.889039	0.967742	
Algorithm	0.962420	0.985180	0.975228	
Education	0.960620	0.960514	0.977802	
Service	0.952022	0.950168	0.973591	
Need	0.952022	0.950168	0.973591	
Material	0.976161	0.982384	0.970806	
Tree	0.961761	0.976401	0.972485	
Rice	0.990378	0.979694	0.978224	
Farmer	0.984688	0.982962	0.985983	
Hospital	0.988463	0.985607	0.978871	
Fruit	0.980640	0.982110	0.973478	
Learn	0.000000	0.963299	0.970944	
For	0.000000	0.918602	0.934661	
Research	0.000000	0.951872	0.945421	
Most	0.000000	0.842863	0.805416	
Average score	0.793520	0.945199	0.961164	
Similarity test without out-of-vocabulary cases	
Student	0.972235	0.969822	0.980334	
Network	0.960879	0.900816	0.949458	
Computer	0.926148	0.876713	0.976250	
Medical	0.947056	0.941894	0.953158	
School	0.955998	0.940223	0.970204	
Architecture	0.940479	0.946469	0.973333	
Artificial	0.871511	0.936170	0.964170	
Science	0.891511	0.935207	0.970364	
Engineering	0.909928	0.904404	0.958697	
Algorithm	0.962420	0.889039	0.975228	
Education	0.960620	0.960514	0.977802	
Service	0.952022	0.950168	0.973591	
Need	0.952022	0.950168	0.973591	
Material	0.976161	0.982384	0.970806	
Tree	0.961761	0.976401	0.972485	
Rice	0.990378	0.979694	0.978224	
Farmer	0.984688	0.982962	0.985983	
Hospital	0.988463	0.985607	0.978871	
Fruit	0.952224	0.950408	0.973478	
Average Score	0.952224	0.950408	0.970575	
Note:

Bold values indicate the highest similarity score for each word pair among the compared models.

The model’s performance on four words could not be determined when considering OOV words, as shown in the tables. Previous studies have reported that the Fast2Vec achieved a mean similarity score of 0.970575, which is significantly higher compared to its alternative models, namely FastText (0.950408) and Word2Vec (0.952224). Fast2Vec consistently outperformed these models and demonstrated varying performance across different word pairs, such as ‘computer’ (0.976250), architecture’ (0.973333), and ‘education’ (0.977802). Further support proves the claim that the semantic quality of embedding is largely influenced by the underlying model architecture. Additionally, word pairs categorized under more abstract or context-dependent terms, such as ‘you’ and ‘others’, exhibited lower semantic consistency across models.

In certain cases, FastText performed comparably to Fast2Vec, particularly for words like ‘material’ (0.982384). However, in majority of instances, Fast2Vec either outperformed or matched the performance of alternative models, highlighting its superior ability to capture nuanced semantic relationships in textual data.

Dimension reduction

To reduce dimensionality, UMAP was employed on the high-dimensional word embeddings obtained from the Fast2Vec model. These embeddings were 300-dimensional and were meant to retain subword and semantic information. The UMAP successfully mapped the high-dimensional spaces into two-dimensional spaces, enabling straightforward representation and grouping of the data. UMAP was preferred to t-SNE because of its ability to retain many data features without being impacted by the “crowding problem,” which tends to alter how the data is represented. Also, the computational time cost of UMAP was much lower than that of t-SNE, thus making it appropriate for the large datasets pertinent to this study.

The algorithm’s effectiveness was supported by observations made during the dimensionality reduction processes since there were distinct clusters of semantically related words. For example, words such as network, node, and graph were included in one group as they were dominantly used in computer science. In contrast, words like education, learning, and teaching formed a separate group around education. This approach would guarantee that critical semantic information was preserved and thus could readily facilitate its appropriate rescaling for further modeling where topical change and semantic features would be investigated.

Semantic clustering

AP was employed as the clustering method for its ability to autonomously determine the number of clusters based on vector similarity, making it particularly suitable for semantic analysis. In initial trials with a vocabulary size of 10,000 words, the algorithm required 10–15 min of runtime and produced one hundred clusters when the maximum iteration was set to one hundred. However, the centroids had not stabilized as data points continued to move between clusters. When the vocabulary size was increased to 55,000 words, AP computed pairwise distances for all vocabulary terms, significantly increasing computational complexity. This underscores the scalability challenge of AP when dealing with large datasets.

Despite these challenges, AP’s strengths lie in its ability to compute a similarity matrix using the negative Euclidean distance between vectors, enhancing the identification of cluster centers (exemplars) that best represent each semantic category. Unlike K-Means, AP does not require the number of clusters to be predefined and exhibits greater flexibility than DBSCAN in managing datasets with diverse distributions. By leveraging preference values derived from the median of the similarity matrix, AP complements the Fast2Vec model by integrating subword information from FastText and semantic relationships from Word2Vec, producing more meaningful and contextually relevant clustering results.

Analysis of evolving dynamics

The topic-semantic probability distribution, represented by θk,ctk emerges from the relationship between topic-word distributions and semantic space. Fluctuations in this distribution indicate how topics evolve, with cosine distance between word vectors preserving conceptual differences (Kim, Kim & Cho, 2017). Semantic evolution is further analyzed using linear regression, where time is the independent variable and topic-semantic probabilities are the dependent variable. Positive coefficients suggest topic concentration in specific semantic ideas, while negative coefficients reflect dispersion across concepts—entropy measures topic concentration, with lower entropy indicating more focused topics. Entropy is calculated using H(x), as described in Eq. (14).

(14) H(x)=−∑i=1n⁡p(θi)log2p(θi).

Integrating UMAP for dimensionality reduction, AP for clustering, and Fast2Vec for enhanced word representation enables more coherent dynamic topic analysis. The approach reveals how topics evolved in semantic spaces, offering insights into their concentration and dispersion over time. It also contributes to a better understanding of topic evolution in scientific research.

As the analysis progresses, the model emphasizes the evolving dynamics of topics and semantics over time. By matching topic words to their corresponding semantic clusters, the model derives topic-semantic probability distributions for different periods. This enables the discovery of evolving dynamics using line charts, which capture changes in topic intensity and semantic relationships year by year. Statistical measures such as entropy and linear coefficient calculations are used to quantify these transformations further. These analyses uncover regular patterns in topic evolution and highlight significant shifts, offering a comprehensive view of how topics and their meanings evolve over time.

Visualization dynamic pattern

By resorting to detailed reporting, it becomes easier to better comprehend the intrinsic role each topic has in academic and practical matters. Each emerging topic is presented alongside river graphs, line charts, bar charts, and stacked area plots, which link to a more extensive discussion of prints and topic semantic change over time. Moreover, these reports give a better understanding of each topic’s emergence, transformation, and interaction with others. All this visualization is synthesized into visual representations. Alongside the preprocessing, topic detection, and semantic analysis, all blend in excellently, giving insights into the modification of semantics and topics contained within large-scale resolution.

Experiments and results

Performance measurement

Perplexity was used as an evaluation metric in this study. The cross-validation technique divided the dataset into three equal parts, and the model was trained and evaluated three times using different fold combinations. In each iteration, one-fold serves as the test data, while the other two are used for training. This approach evaluates how well the model can represent the distribution of words in the test data based on the resulting topic, as Wallach et al. (2009) suggested. The perplexity score reflects the model’s uncertainty in predicting new data, with a lower score indicating better performance in capturing word distribution patterns. This score has also been consistent with other cluster quality metrics (Tang et al., 2014).

The Fast2Vec model integrates FastText and Word2Vec, enabling it to handle OOV issues effectively. As a result, it produces superior semantic representations. Comparative tests using four target words (“student,” “network,” “computer,” and “scrum”) demonstrate the closer similarity between Word2Vec and FastText, as reflected in Table 7 through cosine similarity calculations. The table provides detailed comparisons of semantic associations generated by Word2Vec, FastText, and Fast2Vec, highlighting Fast2Vec’s ability to maintain contextual relevance across diverse word categories.

Table 7 Similarity comparison of Word2Vec, FastText, and Fast2Vec.

Similarity Word2Vec	Similarity FastText	Similarity Fast2Vec	
Student	Network	Computer	Scrum	Student	Network	Computer	Scrum	Student	Network	Computer	Scrum	
teacher	node	software	agile	studentsof	networks	minicomputer	scrub	learner	networkswith	minicomputer	agile	
learner	system	smartphone	plomp	of student	networkbased	microcomputer	agile	teacher	networking	microcomputer	erp	
mathematics	architecture	advanced	waterfall	ustudent	networking	computeraided	developer	studentsof	networked	plc	alessi	
participant	server	internet	prototyping	student	networked	komputer	projectbased	mathematics	networkbased	hardware	project	
teaching	bandwidth	digital	laravel	studens	subnetworks	computerbased	devops	studentteachers	wireless	desktop	software	
lecturer	routing	android	togaf	the students	netowork	computerised	uml	astudents	system	bluetooth	devops	
achievement	networking	plc	sdlc	astudents	mipmanet	computerization	define	ofstudent	subnetworks	device	backlog	
classroom	wireless	desktop	nieveen	ofstudents	hetnet	compute	scrutiny	teaching	bandwidth	software	webbased	
skill	WSN	bluetooth	dick	researchers	hetnets	komputers	dml	participant	vanets	advanced	vasicek	
lesson	grid	Tool	trollip	studets	rbanet	computerized	erp	ustudent	netowork	platform	scm	
writing	UAVs	GPU	devops	stude	fanets	software	project	studens	mipmanet	hmi	multiplatform	
english	connection	smartphones	lewin	studentcentered	node	computerize	del	thestudents	routing	interfacing	webrtc	
academic	computing	camera	alessi	teacher	routing	computes	scr	ofstudents	server	microprocessor	gamified	
reading	qos	web	prometheus	learner	subnets	optisystem	sdlc	stude	protocol	raspberry	workflow	
math	protocol	technology	programming	teaching	subnet	puter	semmel	studentcentered	architecture	smartphone	architecture	
school	device	virtual	eucs	mathematics	manet	softcomputing	webrtc	lecturer	manet	computerised	nosql	
motivation	transmission	engineering	bmst	teacherpreneur	multisystem	multisystem	waterfall	learning	node	application	cloud	
class	vpn	hmi	zachman	learning	fanet	programmable	scroll	child	channel	smartphones	gall	
instructional	link	platform	software	selflearning	multiprotocol	automation	scrofa	english	multipoint	computeraided	scrub	
lecture	router	hardware	uxl	mathemathics	system		sql	writing	computing	zigbee	crawler	

Figure 2 shows the bar chart that illustrates a comparative analysis of cosine similarity scores for the semantic representation of selected words using three embedding models: Word2Vec, FastText, and Fast2Vec. Each bar corresponds to the similarity vector for a specific word, enabling an evaluation of the model’s performance in capturing semantic relationships. Across the dataset, Fast2Vec consistently outperformed Word2Vec and FastText regarding similarity scores, demonstrating its robustness in representing semantic nuances.

Figure 2 Comparing the similarity vector.

Blue color: Word2Vec. Orange color: FastText. Green color: Fast2Vec.

Evaluation on benchmark word similarity datasets

To evaluate the semantic representation capability of Fast2Vec, the model was tested on twelve standardized word similarity benchmark datasets: EN-MC-30, EN-MEN-TR-3k, EN-MTurk-287, EN-MTurk-771, EN-RG-65, EN-RW-Stanford, EN-Simlex-999, EN-Verb-143, EN-WS-353-ALL, EN-WS-353-REL, EN-WS-353-SIM, and EN-YP-130. These datasets capture a range of semantic characteristics, including similarity vs relatedness, syntactic variation, and differences in word frequency. Evaluation was conducted using Spearman’s rank correlation and Pearson’s correlation, which used to assess the relationship between model-generated cosine similarity scores and human-annotated similarity ratings (Finkelstein et al., 2001; Mukaka, 2012; Zhang & Wallace, 2015).

Fast2Vec achieved the highest Spearman correlation in seven out of 12 datasets and the highest Pearson correlation in six out of 12. It performed particularly well in datasets requiring fine-grained semantic understanding (e.g., EN-SimLex-999, EN-WS-353-SIM) and those involving rare or morphologically complex words (e.g., EN-RW-Stanford). The results are presented in Table 8.

Table 8 Spearman and Pearson correlation scores on twelve standard word similarity datasets for Word2Vec, FastText, and Fast2Vec.

Dataset	Word2Vec
(S)	FastText
(S)	Fast2Vec
(S)	Word2Vec
(P)	FastText
(P)	Fast2Vec
(P)	
EN-MC-30	0.4167	0.4602	0.4292	0.4433	0.5030	0.4791	
EN-MEN-TR-3K	0.4420	0.4537	0.4542	0.4606	0.4582	0.4587	
EN-MTurk-287	0.3234	0.3709	0.4011	0.3691	0.4080	0.4131	
EN-MTurk-771	0.4952	0.4420	0.4812	0.5093	0.4448	0.4848	
EN-RG-65.txt	0.3348	0.4264	0.2940	0.3879	0.4000	0.2770	
EN-RW-STANFORD	0.0673	0.2812	0.2784	0.0987	0.2949	0.2922	
EN-SIMLEX-999	0.1835	0.2261	0.2395	0.2053	0.2440	0.2665	
EN-VERB-143	−0.0168	0.1408	0.1622	0.0203	0.2026	0.2190	
EN-WS-353-ALL	0.3626	0.4047	0.4173	0.3972	0.4329	0.4357	
EN-WS-353-REL	0.4119	0.4122	0.4303	0.4272	0.4297	0.4342	
EN-WS-353-SIM	0.3942	0.4665	0.4725	0.4291	0.4995	0.4859	
EN-YP-130	0.2848	0.3472	0.3329	0.2814	0.3609	0.3538	
Note:

S, Spearman; P, Pearson. Values in bold represent the best results achieved for each evaluation metric.

These findings suggest that Fast2Vec produces more consistent and generalizable semantic representations than both Word2Vec and FastText. Its superior performance on linguistically complex datasets demonstrated its robustness in contexts where traditional models tend to underperform. Bu combining the contextual sensitivity of Word2Vec with the subword modeling strengths of FastText, Fast2Vec emerges as a more reliable embedding model for capturing nuanced semantic relationships across diverse linguistic settings.

Topic discovery and distribution

After defining eight topics with DTM, topic probability distributions were computed across the dataset, and representative terms were identified through expert validation. These topics cover key areas such as food availability, performance analysis, health research, and economic stability, offering insight into the dataset’s thematic diversity.

Initially, Topic 1 focuses on agriculture, natural resources, and rural development. Words such as ‘farmer,’ ‘water,’ ‘fiber, and ‘food’ emphasize farming and natural products. In addition, terms like ‘score,’ percentage, and ‘value’ indicate that measurement and evaluation play a role in this topic. This combination points toward discussions involving agricultural productivity, resource management, or rural economic assessments.

Similarly, Topic 2 emphasizes governance, institutional performance, and societal development. Terms like ‘government,’ power,’ and ‘analysis’ suggest a focus on administrative functions and decision-making processes. Moreover, words like ‘design,’ performance,’ and ‘character’ introduce the evaluation of organizational efficiency or institutional frameworks. As a result, this topic likely explores themes of public administration, governance, or strategic planning.

Topic 3 focuses on public health, surveys, and social research. The presence of terms such as ‘health,’ respondents,’ sampling,’ and ‘control’ implies a connection to empirical studies, data collection, or public health initiatives. Additionally, the recurrence of the word “number” highlights the quantitative aspect of this topic. Consequently, this cluster may represent health-related surveys, population studies, or broader social research.

Topic 4 revolves around economic systems, financial institutions, and social structures. Words like ‘finance,’ ‘bank,’ ‘company,’ and ‘local’ indicate a strong focus on economic development and financial operations. Furthermore, terms such as ‘community,’ institution,’ and ‘social’ suggest the interplay between economic growth and societal dynamics. Thus, this topic addresses local governance, economic policies, and community development.

Despite their differences, there are notable overlaps among the topics. For instance, government’ and ‘power’ appear in multiple columns, underscoring the interdisciplinary nature of governance and leadership. This repetition suggests that while the topics diverge in focus, they may collectively contribute to a broader narrative about societal progress, economic growth, and public welfare.

Visualization and topic trend analysis

Figure 3 visualizes the evolution of topic intensity from 2009 to 2023, highlighting fluctuations and trends in the prominence of each topic over time. Topic intensity is calculated based on the normalized distribution of document-topic associations for each year, reflecting the collective attention given to a particular topic within the dataset. These visualizations reveal notable differences in how heavily certain topics were emphasized during specific periods. For example, Topic 1 consistently dominated throughout the years and experience a sharp increase in prominence in 2014, indicating its heightened significance. In contrast, Topic 2 and Topic 3 exhibited moderate and shifting levels of focus, reflecting realignments in research agendas.

Figure 3 Trend Topic.

Blue color: Topic 1. Orange color: Topic 2. Green color: Topic 3. Light blue color: Topic 4. Purple color: Topic 5. Light green color: Topic 6. Dark blue color: Topic 7. Chocolate color: Topic 8.

Topics 6 and 7 showed lower intensities, suggesting they were niche areas or of less popularity. These patterns emphasize the dynamic growth of some topics and decline of others, illustrating the evolving nature of research interests. This comprehensive analysis underscores the importance of employing effective analytical techniques, such as the proposed Fast2Vec integration, to account for the structural changes and temporal dynamics of topics over time.

Figure 4 illustrates changes in topic intensity from 2009 to 2023. Topic 1 (food availability) experienced a sharp peak in 2013–2014, reaching an intensity of 0.43, likely driven by global attention to food crises or agricultural challenges. Afterward, its intensity declined but stabilized between 0.2 and 0.3, remaining relevant throughout 2023. Topic 2 (performance analysis) showed a stable trend throughout the period, with consistent intensity levels between 0.2 and 0.3, reflecting sustained interest. Topic 3 (health and epidemiology) remained steady, with minor fluctuations and a slight increase approaching 2023, likely tied to rising interest in global health issues.

Figure 4 Topic size.

Red color: Topic 1. Light blue color: Topic 2. Yellow color: Topic 3. Light green color: Topic 4. Dark purple color: Topic 5. Chocolate color: Topic 6. Light purple color: Topic 7. Dark green color: Topic 8.

Topic 4 demonstrated relatively stable intensity between 0.1 and 0.25 without significant spikes. Topic 5 saw a notable rise at the start of the period but dropped sharply around 2015, suggesting temporary attention to forest ecosystem management. Topic 6 showed moderate fluctuations, with intensity ranging between 0.1 and 0.2, indicating a steady focus on research methodologies. Topic 7 (health and biomedical research) maintained low intensity over time, with a slight rise in 2018. Finally, Topic 8 (banking business) exhibited a steady increase, peaking in 2016, followed by a decline until 2023.

Overall, Topic 1 stands out with a significant peak in 2013–2014, while Topic 2 and Topic 4 displayed strong stability. Topic 8 showed an early rise but experienced a notable decline after its peak in 2016.

Evolution of dynamics in semantic space

Figure 5 shows the clusters representing various semantic categories connected to each topic. The evolution of the eight topics over time demonstrates distinct patterns of dominance and fluctuations across clusters. In Cluster 1, the dominant topic was Topic 1 (food availability), which peaked around 2014 and 2022, reflecting heightened discussions on local agricultural products and water availability, likely in response to global or regional agrarian challenges. Topic 2 (performance analysis) went through significant changes (especially in Cluster 4) and peaked in 2015 and 2018. This suggests that performance evaluations were more important during improved governance and operation. For Topic 3 (health and epidemiology), Cluster 1 consistently dominated, peaking in 2021, possibly due to the global focus on public health issues during the COVID-19 pandemic.

Figure 5 The eight main key topic.

(A) Food availability. (B) Performance analysis. (C) Health and epidemiology. (D) Economic stability. (E) Foreign ecosystem management. (F) Research methodology. (G) Health and biomedicine. (H) Banking and business.

Topic 4 (economic stability) showed relative stability in Cluster 4, with peaks in 2018 and 2023, highlighting ongoing attention to financial and governance-related topics. Meanwhile, Topic 5 (environmental management) saw a decline in Cluster 1’s dominance, replaced by a significant spike in Cluster 8 in 2023, indicating a shift toward algorithmic and technical approaches to environmental research. Topic 6 (methodological research) demonstrated a change in Cluster 1, peaking in 2023, suggesting the growing relevance of statistical methods like correlation and sampling. In Topic 7 (biomedical research), Cluster 1 maintained dominance, with Cluster 10 peaking in 2018, reflecting research on public health and cholesterol-related studies. Finally, Topic 8 (Islamic banking) showed a decline in Cluster 1 over time, with a sharp increase in Cluster 4 around 2018, highlighting the periodic focus on Islamic banking management.

Overall, the analysis underscores how global, regional, and disciplinary dynamics influence the evolution of research priorities across topics.

Analysis of evolving dynamic topic patterns

Entropy measures the dispersion or concentration of a topic across different clusters. A topic spreads across multiple clusters, indicating a wide variation or diffusion in its semantic focus when its entropy value is closer to 1.3. Conversely, a lower entropy value (closer to −0.1) suggests a more concentrated focus within a limited number of clusters. Figure 6 indicates peaks and valleys in entropy for the eight topics over time. These variations reflect how each topic’s relevance and focus changed, either by becoming more diffuse (spread across clusters) or concentrated (focused on fewer semantic areas).

Figure 6 Three-dimension surface plot illustrating the temporal entropy values of topic T_1 to T_8 over the publication period from 2009 to 2023.

Entropy represents the degree of uncertainty or dispersion of each topic across time, where higher values indicate broader diffusion or instability. The vertical axis indicates the entropy value, while the horizontal axes correspond to publication year and topic index. Color variations highlight the dynamic changes in topic distribution over time.

Topic 1: Entropy fluctuates, showing moments of both concentration and diffusion. Around 2014, entropy peaks, indicating a more varied discussion across several semantic clusters, due to diverse research interests during that period. Topic 2 (performance analysis): There was a noticeable peak in entropy around 2018, indicating that discussions on performance analysis were spread across multiple clusters, suggesting varied approaches or subtopics gaining attention simultaneously. Topic 4 (economic stability): This topic showed several spikes in entropy, particularly in recent years (2020–2023), indicating that economic discussions became more fragmented, reflecting the broadening scope of financial stability to cover different aspects like governance, finance, and societal impact.

Topic 7: Entropy remains low, suggesting that interest in this topic stayed more focused over time, concentrating on specific subtopics, such as biomedical interventions or health metrics, with less diffusion across other semantic clusters.

Diffusion vs. stability: trend insights. Topics such as Topic 1 and Topic 2 exhibited high entropy at different points, indicating periods of exploration across multiple dimensions or subtopics. On the other hand, Topic 7 maintained a low entropy trend, suggesting that the topic stayed consistently focused on a core set of ideas or clusters. Significant shifts: Sudden jumps or dips in entropy, such as the peaks seen for Topic 6 and Topic 5, indicate periods where these topics either expanded in scope or became more specialized. These shifts are crucial for understanding how research trends fluctuate over time.

The overall entropy distribution suggests a mix of diffusion, where topics spread into new semantic areas, and stabilization, where specific topics retain focus over time. The graph provides a visual illustration of the dynamic interplay between these modes, helping to track how topics evolve and shift in relevance across years.

Both Figs. 5 and 6 illustrate the semantic evolution of eight topics over time (trends), highlighting the dynamic shifts in their probability distributions. Each topic exhibits unique patterns of fluctuation, revealing periods of heightened focus and diversification. Peaks in the graph signify moments of expansion where research broadened to include various subtopics, aligning with increased entropy values. Conversely, lower-intensity periods indicate a concentrated focus on specific thematic areas, reflected in reduced entropy values.

These dynamic transitions underscore the varying nature of topic evolution, showing how some topics, such as Topics 1 and 5, maintain consistent prominence, while others, like Topics 3 and 7, exhibit intermittent surges. Clustering within each topic provides additional insights into the interrelation and differentiation of subtopics over time. This evolving semantic landscape demonstrates the complexity and fluidity of research focus, offering valuable perspectives for understanding the shifts in academic priorities and thematic structures over the analyzed period. These results show how important it is to use useful models, like Fast2Vec, to understand these complex changes and give accurate explanations of how topic structures are changing.

Discussion

This study, which integrated the Fast2Vec model with dimensionality reduction and clustering techniques, provides significant insights into the evolution of research topics between 2009 and 2023. This section will discuss our findings about categorized trends (diffusion, stability, moderate fluctuation, and shift), entropy-based dynamic, and the evaluation of semantic representation performance.

Semantic evolution and topic dynamics

The categorization of diffusion, stability, moderate fluctuation, and shift topics accurately reflects the dynamic nature of research development, as shown in Fig. 7. Topics like food availability and performance analysis demonstrate the broadening of the research scope over time, as indicated by high entropy values. These diffusion topics suggest increasing interdisciplinarity and incorporating new research dimensions, driven by global challenges such as food security and performance evaluation in governance and education.

Figure 7 Temporal entropy trends across four representative topics from 2009 to 2023.

(A) Diffusive topics. (B) Stable topics. (C) Drift topics. (D) Moderate fluctuation topics. Each subplot (A–D) illustrates entropy dynamics, highlighting patterns of diffusion, stability, drift or fluctuation moderate topic distributions over time.

In contrast, stable topics such as health and biomedical research exhibit concentrated focus, due to continuous interest in specific subtopics, particularly during major health crises. Low entropy value for these topics suggests a lack of significant semantic shifts over time.

Shifts and fluctuations in research focus

Moderate fluctuation topics, such as forest ecosystem management, show periodic intensity spikes, often in response to external events or new technologies. Though dynamic, these topics maintain an overall steady trajectory. Shifts topic like economic stability demonstrates significant changes across clusters over time, often triggered by socio-political and economic events, as reflected in increasing entropy in later years.

Fast2Vec model and semantic clustering

By combining subword information from FastText with contextual sensitivity from Word2Vec, Fast2Vec effectively capture nuances semantic variation within topic clusters. The use of UMAP and AP further supports the temporal interpretation of topic structures. These results confirm Fast2Vec’s utility for semantic-based topic modeling, particularly in handling OOV terms.

Evaluation of semantic similarity

The findings presented in Table 8 provide strong empirical support for the semantic reliability of the Fast2Vec model. Fast2Vec consistently performed well across twelve word similarity datasets, covering diverse linguistic features such as similarity vs. relatedness, Part-of-Speech (POS) categories, and term frequency. It achieved the highest Spearman correlation in seven datasets and the highest Pearson correlation in six, indicating strong alignment with human-annotated similarity judgments.

Fast2Vec outperformed both Word2Vec and FastText on several challenging benchmarks. These include EN-Simlex-999, EN-RW-Stanford, and EN-Verb-143. Notably, Word2Vec yielded a negative Spearman correlation (−0.0168) on the EN-Verb-143 dataset, underscoring its difficulty in modeling syntactic variation, an area where Fast2Vec demonstrated clear robustness. While FastText showed strengths in capturing subword-level patterns, it occasionally lacked contextual coherence. conversely, Word2vec remained effective on lexical stable datasets, such as EN-MEN-TR-3k and EN-YP-130, but struggled with infrequent or morphologically rich terms.

Overall, the results underscore Fast2Vec’s ability to integrate subword modeling and contextual semantic representation. This integration produce robust and consistent outcome in word similarity tasks. These findings position Fast2Vec as a promising hybrid embedding model capable of capturing nuanced semantic relationship across diverse linguistic settings.

Implication of entropy trends

Entropy, as a measure of topic dispersion or concentration, provided more profound insights into how topics evolved in semantic space. Topics with high entropy values are indicated to be widespread across semantic clusters, reflecting a diffusion of ideas or the emergence of new subtopics. On the other hand, low entropy values highlighted concentrated areas of research where topics remained focused on specific semantic themes. Particularly in topics like performance analysis, the peaks in entropy indicate periods of simultaneous pursuit of multiple research directions. This spread may reflect increased interdisciplinary collaboration or diversification of methods and approaches within the topic.

Broader research implications

The topic categorization reveals that diffusion topics reflect the need for adaptable research paradigms, while stable topics offer essential continuity. Shift and diffusion topics highlight responsiveness to external stimuli, guiding future prioritization. The combines semantic evaluation and topic dynamics analysis affirm Fast2Vec’s role as a powerful model for exploring and interpreting topic evolution over time and contributes both to the refinement of topic modeling frameworks and the advancement of hybrid word embedding techniques for semantic analysis.

These findings align with prior research, such as Zhang et al. (2019), which proposed a generative model to detect dynamic topical communities by integrating topic evolution with author networks. While their approach emphasized structural and probabilistic modeling, the present study offers a complementary perspective by focusing on semantic representation using hybrid embeddings and entropy-based topic categorization. This integrative strategy enhances interpretability and validates topic evolution with quantitative semantic similarity benchmarks aligned to human judgments.

Conclusion

The evaluation results demonstrate that the Fast2Vec model significantly outperforms Word2Vec and FastText in representing semantic similarity across diverse linguistic scenarios. In OOV setting, Fast2Vec showed a 39.64% improvement over Word2vec and a 6.18% improvement over FastText. In non-OOV cases, it still surpassed FastText by 7.82% and Word2Vec by 0.087%. Additional experiments using twelve widely recognized word similarity benchmarks further validated its consistency and robustness. Fast2Vec achieved the highest Spearman correlation in seven datasets and the highest Pearson correlation in six, particularly excelling in tasks involving rare or morphologically complex terms.

By combining the contextual strength of Word2Vec with the subword modeling capability of FastText, Fast2Vec addresses long-standing limitations in word embeddings (Bojanowski et al., 2017; Mikolov et al.), but also highlight Fast2Vec practical potential in capturing nuanced semantic relationship across tasks.

In addition to semantic evaluation, the model was applied to a large-scale topic modeling task using research abstracts from the SINTA database. Eight dominant research topics were identified and categorized into four dynamic patterns: diffusion, shift, moderate fluctuation, and stability. These classifications were supported by entropy-based analysis and are consistent with established frameworks on topic evaluation and innovation diffusion (Rogers & Everett, 2003; Blei & Lafferty, 2006).

Future research could explore (1) applying Fast2Vec to author network analysis to uncover semantic-driven shifts in research collaboration; (2) refining the handling of OOV terms through subword tuning or hybrid models; (3) implementing advanced text preprocessing to reduce semantic noise; (4) extending evaluations across diverse datasets and languages to assess domain adaptability; and (5) releasing Fast2Vec as an open-source tool to encourage broader adoption and experimentation in the NLP community.

In summary, Fast2Vec contributes significantly to the advancement of word embedding techniques by enhancing semantic precision and generalizability. It serves as a strong foundation for dynamic topic modeling and other semantic-rich NLP tasks. With its demonstrated empirical strengths and flexibility, Fast2Vec offers promising potential for future research and practical applications in interpretable and context-aware language modeling.

Additional Information and Declarations

Competing Interests

The authors declare that they have no competing interests.

Author Contributions

Ayu Pertiwi performed the experiments, analyzed the data, performed the computation work, prepared figures and/or tables, authored or reviewed drafts of the article, and approved the final draft.

Azhari Azhari conceived and designed the experiments, analyzed the data, performed the computation work, authored or reviewed drafts of the article, and approved the final draft.

Sri Mulyana performed the computation work, prepared figures and/or tables, and approved the final draft.

Data Availability

The following information was supplied regarding data availability:

The code is available at Zenodo: Ayu Pertiwi. (2025). Fast2Vec: A Modular Framework for Semantic Analysis and Topic Modeling. Zenodo. https://doi.org/10.5281/zenodo.14613327.

The dataset is available at Zenodo: Ayu Pertiwi, A. P. (2024). SINTA Journal [Data set]. Zenodo. https://doi.org/10.5281/zenodo.14613315.

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
