# Peer review of "Fast2Vec, a modified model of FastText that enhances semantic analysis in topic evolution"

_PeerJ Computer Science, doi:10.7717/peerj-cs.2862_

## Round 0.1 · original submission · Major Revisions

Dear Authors,

Thank you for submitting your article. Reviewers have now commented on your article and suggest major revisions. We strongly recommend that you address the issues raised by the reviewers, especially those related to readability, experimental design and validity, and resubmit your paper after making the necessary changes.

Warm regards,

Reviewer 1 ·

Basic reporting

1. There are many embedding methods in the literature, such as BERT family. This work does not discuss such related work.

2. There are many typos in the mathmatical expressions, for examples in line 264, 280,282,308, 311. There are Typos in equations, like eq.(4). The mathematical presentations are poor.

Experimental design

In algorithim4, you defined the evaluation metric of cosine similarity of word pairs. Still, there is no description of picking the word pairs and no comparison of the performance of FastText and Word2Vec.

Validity of the findings

This work proposes to improve the word2Vec algorithm by merging the FastText algorithm; however, it does not compare with other popular tokenizer and embedding models, such as Bert. This work is not well-motivated.

Additional comments

The analysis section is comprehensive, but the motivation behind the work feels relatively weak. Additionally, the presentation of the mathematical functions could be improved.

Reviewer 2 ·

Basic reporting

1. Equation. 1 is a little bit confused. Please explain more about the meaning of the symbols. For example, the meaning of (w1)^n, and the i in the (wi)^n from the previous sentence.

2. There are many typos, especially in equations. I suggest the authors review all the equations and tables carefully to fix them. Below are some of them.
(1) Typo in Table.1 “Document 54.689” should be “Document 54,689”.
(2) In the “Semantic Analysis” section, the “Yword” should be in formula format.
(3) Missing “=” in Equation. 4.
(4) Missing one “(” in Equation 5 and 6.
(5) There is a “;” in Equation 8.

3. It would be better to add detailed explanation or description for each table and figure.

4. The organization of the article is a little confusing. Background and methods should be clearly separated, and each part of the method should be presented in a separate section rather than mixed together. For example, the whole pipeline seems to be summarized again in the "Out-of-Vocabulary (OOV) Handling" section.

Experimental design

1. In the “Topic Detection” section, the evaluation of the ideal number of topics is not clear. I am confused that why eight topics were the ideal number. May include tables to show the difference between the selected topics and ignored topics like Table 6.

2. How did you decide the parameters of the model? Especially for the weights to combine the Word2Vec model and the FastText model. It would also be better to mention the exact parameters for better reproduction.

3. There are many other word dimensionality reduction methods, such as t-SNE. I am wondering why you chose UMAP for dimensionality reduction. Adding some explanation would be better.

4. There are many other clustering methods, such as DBSCAN. I am wondering why you chose AP for clustering. Adding some explanation would be better.

Validity of the findings

1. In terms of comparative tests in the “Performance Measurement”, examples in Table 4 are not convinced enough since most words are similar to the given word. Including more obvious examples would be better.

2. Lack of explanation of exact evaluation process of Table 5. I am confused as to why there are only 14 words and 10 words shown in Table 5a and 5b, respectively. The table will not be convincing enough with so few words calculated.

3. Figures need more explanation. Figure 3 and Figure 5 have the same X and Y axis but look different. It is hard to understand the difference between them.

4. Figure 8 is missing some legends.

Additional comments

The paper’s idea of combining Word2Vec and FastText models and demonstrating the evolution of research topics from 2009 to 2023 is interesting. However, the paper has many problems, and the numeric results demonstrating the effectiveness of the Fast2Vec model are not convincing enough. The authors need to improve the paper carefully to be accepted.

Reviewer 3 ·

Basic reporting

The document is written in professional English and provides a clear introduction and background. However, there are instances where sentence structure is overly complex, which may affect readability. Simplify some of the sentences to enhance clarity for a broader audience. For example, the sentence in the introduction, "Without proper analysis, scientific data will remain meaningless raw information," could be rephrased as, "Proper analysis is essential to transform raw scientific data into meaningful insights."

Experimental design

The experimental design is described in detail, and the methods are replicable. However, there is limited explanation of how the dataset was cleaned and pre-processed before training models.
Improvement: Include a more detailed explanation of the data pre-processing steps, such as handling missing values, removing duplicates, or normalizing text, to provide better transparency and replicability.

Validity of the findings

-The paper does not adequately discuss the limitations of the proposed model. For example, the potential bias in the dataset (limited to SINTA journals) or the generalizability of Fast2Vec to other languages or domains needs to be addressed.

-The findings are well-supported by statistical evidence, and the improvements of Fast2Vec over Word2Vec and FastText are clearly quantified. However, the document does not address potential limitations or biases in the dataset or method.
Improvement: Add a section discussing the limitations of the study, such as potential biases in the SINTA dataset (e.g., specific domains being overrepresented) or the generalizability of Fast2Vec to other datasets or languages.

Additional comments

- The semantic analysis section effectively explains the integration of Fast2Vec with LDA, but it lacks examples of how the grouping of words into semantic concepts improved interpretability.
- The description of Fast2Vec's architecture and functionality is comprehensive, but the explanation of how the weights for merging Word2Vec and FastText results were determined is unclear.
- The conclusion summarizes the contributions of the study well but does not propose specific future research directions.

---

## Round 0.2 · Minor Revisions

Dear Authors,

Two of the reviewers did not respond to the invitation to review within the designated timeframe. A new round of reviews is therefore required following the completion of a revised manuscript, which has undergone minor edits in accordance with the comments provided by the reviewer, as outlined below.

Best wishes,

Reviewer 2 ·

Basic reporting

1. The article seems to have some duplicate content. For example, the statements of Hu et al. (2017) and (Gao et al., 2022) appear twice in the Introduction section.
2. Miss a “-” for equation (12).
3. There is a lot of misuse of “.” and “,” in tables.
4. Word-pair should have two words, but Table 6 only shows one word for each row. It would be better to change the title to “target words” if they are.

Experimental design

1. It would be better to explain how to get the similarity scores for the words in Table 6. I wonder if each word has only one pair or if the results are aggregations from many pairs.

Validity of the findings

no comment

Additional comments

The paper’s idea of combining Word2Vec and FastText models and demonstrating the evolution of research topics from 2009 to 2023 is interesting. The paper's quality improves a lot after the author's revision. But may still need some small revisions.

---

## Round 0.3 · Minor Revisions

Dear Authors,

Reviewer2 has still some minor concerns and suggestions for the quality of the paper. We do encourage you to address these concerns and criticisms and resubmit your article once you have updated it accordingly.

Best wishes,

Reviewer 2 ·

Basic reporting

1. Still typos in the paper. For example, "Cosinus Similarity" in Table 6 should be "Cosine Similarity".

Experimental design

no comment

Validity of the findings

1. The cosine similarity is calculated from two words (A and B). However, in Table 6, there is only one word for one row. If the scores represent individual word pairs, there should be two words (A and B) for each row. If the cosine similarity scores are based on word groups from Table 7, you may state it and explain how you aggregate the word pairs to get the scores.

2. Table 6 may show some good examples of how Fast2Vec can be better than Word2vec and Fasttext. However, the samples are still too few and not convincing enough. Adding experiments on word similarity datasets including EN-MC-30, EN-MEN-TR-3k, EN-MTurk-287, EN-MTurk-771, EN-RG-65.txt ,EN-RW-STANFORD, EN-SIMLEX-999, EN-VERB-143, EN-WS-353-ALL, EN-WS-353-REL, EN-WS-353-SIM, and EN-YP-130 and show the numerical results will offer more convincible results.

3. The discussion of topic trend analysis is good. However, it seems hard to verify the results. I am not sure if there is any other similar work to compare.

Additional comments

The paper’s idea of combining Word2Vec and FastText models and demonstrating the evolution of research topics from 2009 to 2023 is interesting. However, the experimental results in which the author only uses examples to demonstrate the effectiveness are not convincing enough. Testing on some word pair similarity datasets would be better.

---

## Round 0.4 · accepted · Accept

Dear Authors,

Thank you for addressing the reviewers' comments. Your manuscript now seems sufficiently improved and ready for publication.

Best wishes,

Reviewer 2 ·

Basic reporting

no comment

Experimental design

no comment

Validity of the findings

no comment

Additional comments

The paper’s idea of combining Word2Vec and FastText models and demonstrating the evolution of research topics from 2009 to 2023 is interesting. The paper's quality improves a lot after the author's careful revision. I have no more comments.